Exploring the chemical space of influenza neuraminidase inhibitors

Anuwongcharoen Nuttapat 1 2
Shoombuatong Watshara 1
Tantimongcolwat Tanawut 3
Prachayasittikul Virapong 2
Nantasenamat Chanin 1 chanin.nan@mahidol.ac.th
1 Center of Data Mining and Biomedical Informatics, Faculty of Medical Technology, Mahidol University , Bangkok , Thailand
2 Department of Clinical Microbiology and Applied Technology, Faculty of Medical Technology, Mahidol University , Bangkok , Thailand
3 Center for Research and Innovation, Faculty of Medical Technology, Mahidol University , Bangkok , Thailand
Silva Pedro
Electronic publication date: 2016 Apr 19
Publication date: 2016
Volume: 4
Electronic Location ID: e1958
Received 2015 Nov 30; Accepted 2016 Mar 31
Copyright: ©2016 Anuwongcharoen et al.
Copyright year: 2016
Copyright holder: Anuwongcharoen et al.
License: This is an open access article distributed under the terms of the Creative Commons Attribution License, which permits unrestricted use, distribution, reproduction and adaptation in any medium and for any purpose provided that it is properly attributed. For attribution, the original author(s), title, publication source (PeerJ) and either DOI or URL of the article must be cited.
License URL: https://creativecommons.org/licenses/by/4.0/

Keywords: Influenza, Neuraminidase, Neuraminidase inhibitor, Chemical space, QSAR, Scaffold analysis, Molecular docking, Fragment analysis, Data mining, Combinatorial library enumeration

Funding: Goal-Oriented Research Grant E09/2557 Swedish Research Links program C0610701 Office of Higher Education Commission and Mahidol University under the National Research Universities Initiative This work is supported by the Goal-Oriented Research Grant (No. E09/2557) from Mahidol University, the Swedish Research Links program (no. C0610701) from the Swedish Research Council as well as partial support from the Office of Higher Education Commission and Mahidol University under the National Research Universities Initiative. The funders had no role in study design, data collection and analysis, decision to publish, or preparation of the manuscript.

==============================
The fight against the emergence of mutant influenza strains has led to the screening of an increasing number of compounds for inhibitory activity against influenza neuraminidase. This study explores the chemical space of neuraminidase inhibitors (NAIs), which provides an opportunity to obtain further molecular insights regarding the underlying basis of their bioactivity. In particular, a large set of 347 and 175 NAIs against influenza A and B, respectively, was compiled from the literature. Molecular and quantum chemical descriptors were obtained from low-energy conformational structures geometrically optimized at the PM6 level. The bioactivities of NAIs were classified as active or inactive according to their half maximum inhibitory concentration (IC50) value in which IC50 < 1µM and ≥ 10µM were defined as active and inactive compounds, respectively. Interpretable decision rules were derived from a quantitative structure–activity relationship (QSAR) model established using a set of substructure descriptors via decision tree analysis. Univariate analysis, feature importance analysis from decision tree modeling and molecular scaffold analysis were performed on both data sets for discriminating important structural features amongst active and inactive NAIs. Good predictive performance was achieved as deduced from accuracy and Matthews correlation coefficient values in excess of 81% and 0.58, respectively, for both influenza A and B NAIs. Furthermore, molecular docking was employed to investigate the binding modes and their moiety preferences of active NAIs against both influenza A and B neuraminidases. Moreover, novel NAIs with robust binding fitness towards influenza A and B neuraminidase were generated via combinatorial library enumeration and their binding fitness was on par or better than FDA-approved drugs. The results from this study are anticipated to be beneficial for guiding the rational drug design of novel NAIs for treating influenza infections.

Introduction

Influenza is a fatal disease of global public health concern. It is caused by influenza viruses which are envelope segmented-RNA viruses belonging to the Orthomyxoviridae family. The global estimate for seasonal influenza infection is as high as 1 billion cases per year in which approximately 3–5 million cases often develop progressive and severe illness leading to 250,000–500,000 fatalities per year worldwide (World Health Organization, 2014). Among the severe cases, high fatality rates are observed particularly in very young children and elderly people >65 years of age who are considered to be a risk group vulnerable to influenza infection. Thus, influenza infections significantly increase the number of hospitalizations, lead to substantial economical losses from disease intervention, and impact the productivity of society (Peasah et al., 2013).

The current strategy for treating influenza focuses on inhibiting the function of neuraminidase, which is an enveloped enzyme located on the surface of both influenza A and B. Influenza neuraminidase is an exosialidase that recognizes the α-ketosidic linkage between neuraminic acid (or sialic acid) and carbohydrate residues (Von Itzstein, 2011). The influenza virus requires this enzyme to facilitate viral budding of progeny virions out of the cells and to prevent viral aggregation of virus particles. The interaction allows the mature virus to detach from the host cell, resulting in the release of progeny virions from the surface of the host cell. Moreover, neuraminidase also plays a role in the cleavage of neuraminic acid of mucin inside the respiratory tract, thereby facilitating the movement of the virus toward its target cells (Shtyrya, Mochalova & Bovin, 2009). Thus, neuraminidase is a crucial enzyme that facilitates viral spreading and transmission. To prevent the spreading of influenza viruses, neuraminidase inhibitors (NAIs) are currently an effective choice for treatment and prophylaxis.

Currently, only three NAIs have been approved for use as therapeutic and prophylaxis agents of influenza virus: zanamivir (Relenza), oseltamivir (Tamiflu) and peramivir (Rapivab). Zanamivir is the first approved nasally administered NAI and it exerts highly effective inhibitory activity against both types of influenza virus. This dihydropyran-based NAI was developed based on the structural modification of a sialic acid analogue called DANA (Meindl et al., 1974). Due to its high polarity, zanamivir exhibits low oral bioavailability and requires administration via nasal inhalation. Oseltamivir is a second-generation NAI approved for use as an oral anti-influenza agent and it exhibits efficacy comparable to that of zanamivir (Tuna, Karabay & Yahyaoglu, 2012). This cyclohexene-based NAI is less polar than the previous generation, thus making it easier to administer than the inhalation route. The most recently approved intravenous NAI, peramivir, was announced in December 2014. This intravenous formulation was developed as a single dose for the treatment of acute uncomplicated influenza infection and it potentially reduces the duration of illness in participants. Although current NAIs exhibit high therapeutic efficacy against circulating influenza virus, searching for novel anti-influenza agents is continuously performed to address newly emerging or mutant strains with resistance to anti-influenza agents.

Nevertheless, a number of drug candidates have failed in the late stages of the drug development process, primarily during clinical trials. These failures are a result of either insufficient therapeutic efficacy or adverse drug reactions at therapeutic doses. Balancing between favorable bioactivity and desirable adverse effects is essential for improving the therapeutic outcome after treatment (Greene & Naven, 2009). The bioactivity of compounds is facilitated by interactions between functional groups aligning inside the molecule and target residues in the binding pocket of the drug target. Thus, insights into the structure–activity relationship are important for filling the knowledge gap during the lead optimization process. Currently, advanced computational-aided drug design approaches are employed in medicinal chemistry research, which potentially reduce costs and the amount of time spent for optimizing a set of novel compounds for pre-clinical and clinical assessments. Chemical space exploration enables the determination of important molecular substructures that contribute to bioactivity against drug targets. In combination with quantitative structure–activity relationships, the informative physicochemical properties and molecular features that are relevant to the bioactivity of compounds can be obtained for discriminating between active and inactive compounds through various machine-learning approaches.

To reduce the failure rate in the late stages of drug design and development, it is necessary to understand both important molecular substructures and informative molecular features relevant to the activity of interest. Herein, we report the application of chemical space for exploring the important structure distributions related to neuraminidase inhibitor activities and the creation of a set of simple physicochemical properties that define the preferred physicochemical properties for neuraminidase inhibition. To achieve this objective, a large data set of neuraminidase inhibitors was collected from a publicly available database of protein-ligand interaction (Liu et al., 2007). This data set provides considerable opportunity for investigating the fundamental profiles that dominate neuraminidase inhibition. Ligand-based approaches, namely univariate, multivariate and scaffold analyses, were performed on compounds in the data set as to explore the chemical space of NAIs. Furthermore, structure-based approaches, namely molecular docking and combinatorial library enumeration, were carried out to generate ligand candidates against neuraminidase from influenza types A and B. Finally, post-filtering of the enumerated ligands were performing using rules from the decision tree model in order to enrich the resulting ligands.

Materials and Methods

Data collection

A schematic workflow is presented in Fig. 1. Bioactive compounds that exhibited an inhibitory effect against neuraminidase of both influenza virus type A and B were collected from BindingDB (Liu et al., 2007), which was primarily compiled from 27 original articles. The bioactivities of the NAIs were indicated by IC50 and converted to pIC50 by taking the negative logarithm to the base of 10 using the following equation: (1) pIC50=−logIC50.

Figure 1 Schematic workflow of this study.

We first excluded the compounds with similar compound names, SMILES structures and protein targets to avoid bias in the prediction model. After the pre-preprocessing procedure, non-redundant data sets consisting of 347 and 175 NAIs for influenza A and B, respectively, were obtained. To categorize compounds as active or inactive, pIC50 cut-off values were used, in which compounds with pIC50 values of greater than 6 (corresponding to an IC50 value of less than or equal to 1 uM) were categorized as “active” and compounds with pIC50 values of less than 5 (corresponding to an IC50 value of greater than or equal to 10 uM) were categorized as “inactive”. Moreover, the intermediate biological activity NAIs with pIC50 values ranging between 5 and 6 were not selected in this study, which consist of 62 and 44 NAIs for influenza virus type A and B, respectively. Finally, sets of non-redundant compounds consisting of 285 influenza A NAIs and 131 influenza B NAIs were obtained and subjected to further investigation. These data sets are provided as supplementary data on figshare and is accessible at http://dx.doi.org/10.6084/m9.figshare.1612484.

Molecular descriptor generation

A molecular descriptor is a numerical description that represents the physicochemical properties and chemical information of compounds. The chemical structures of curated NAIs in the form of SMILES structures were converted to 3D structures using MolConverter from ChemAxon (version 15.1.12.0; ChemAxon Kft., Budapest, Hungary) and then subsequently converted to Gaussian input file format using Open Babel (O’Boyle et al., 2011). Geometrical optimization was performed using density functional theory (DFT) calculations at the PM6 level as implemented in Gaussian09 (Frisch et al., 2009). In this study, low-energy conformations obtained from geometrical optimizations were used to extract thirteen easy-to-interpret molecular descriptors, consisting of six quantum chemical descriptors and seven molecular descriptors, accounting for the physicochemical properties of compounds according to our previous study (Nantasenamat et al., 2013). The obtained quantum chemical descriptors include the mean absolute charge (Qm), energy (E), dipole moment (μ), highest occupied molecular orbital (HOMO), lowest unoccupied molecular orbital (LUMO) and energy gap of the HOMO and LUMO state (HOMO-LUMO). Furthermore, the second sets of molecular descriptors were calculated using DRAGON 5.5 Professional (version 5.5.; Talete, Milan, Italy). The obtained descriptors include the molecular weight (MW), rotatable bond number (RBN), number of rings (nCIC), number of hydrogen bond donors (nHDon), number of hydrogen bond acceptors (nHAcc), Ghose-Crippen octanol-water partition coefficient (ALogP) and topological polar surface area (TPSA). In addition, sets of 307 substructure fingerprint counts (SubFPC) were generated to construct the predictive models of influenza A and B NAIs using PaDEL-Descriptor (Yap, 2011).

Univariate analysis

As an exploratory data analysis, univariate statistical analysis was performed to investigate the different patterns and trends of individual molecular descriptors between active and inactive NAIs using 6 descriptive statistical parameters: the minimum (Min), first quartile (Q1), median, mean, third quartile (Q3) and maximum (Max). In addition, statistical differences of descriptors among active and inactive NAIs were evaluated using the p value obtained from Student’s t-test (Goodman, 1999). Finally, histogram plots of the thirteen descriptors were generated using in-house R language scripts to visualize the different distributions of active and inactive NAIs. The t-test is considered to be a feature selection technique belonging to the class of filter methods (Saeys, Inza & Larrañaga, 2007). The advantages of such filter methods are its fast and scalable nature as well as its independence of the classifier.

Data splitting

The aforementioned non-redundant data sets were divided into internal and external sets with the Kennard-Stone sampling algorithm (Stevens, 2014) using ratios of 80% and 20%, respectively (Table 1). The internal set was subjected to full training calculations and was evaluated using a ten-fold cross-validation (10-fold CV) scheme, which was applied to confirm the reliability and robustness of the proposed models. Furthermore, the external set was used to assess the generalization ability of the model when extrapolating to unknown data samples.

Table 1 Summary of the data set used for predicting the inhibitory activity of influenza A and B.

Data set	Initial	Internal data set	External data set	
		Active	Inactive	Active	Inactive	
Influenza A	285	118	110	30	27	
Influenza B	131	36	69	9	17	

Multivariate analysis

Principal component analysis (PCA) is a tool used for analyzing data sets that possess several inter-correlated quantitative dependent variables (Prachayasittikul et al., 2015; Jolliffe, 2005). To manipulate these inter-correlated variables, PCA essentially transforms the original data into a number of principal components (PCs) or new co-ordinate axes, where the axes are located on the center of the data points. Mathematically, PCs depends on the eigenvectors and eigenvalues of a data covariance (or correlation) matrix. The eigenvector associated with the largest eigenvalue has a direction that is identical to the first principal component (PC1), whereas the eigenvector associated with the second largest eigenvalue determines the direction of the second principal component (PC2) and so forth. In the present study, PCA was employed in exploring the chemical space of NAIs from influenza A and B as a function of the thirteen molecular descriptors using the FactoMineR (Lê, Josse & Husson, 2008) package of the R statistical language. Prior to PCA analysis, all data were first standardized to a comparable scale by transforming variables to zero mean and unit variance.

Decision tree (DT) is a simple, transparent and interpretable learning method that produces decision rules for the underlying data (Quinlan, 1993). Practically, the prediction task using the decision model can be easily implemented without complicated computations and this model can also be applied in both continuous and categorical variables (Prachayasittikul et al., 2015). This algorithm has been widely used for the interpretable analysis of various tasks, such as hepatitis virus C NS5B polymerase (Nantasenamat, Isarankura-Na-Ayudhya & Prachayasittikul, 2010), aromatase inhibitors (Nantasenamat et al., 2013; Shoombuatong et al., 2015b), dipeptidyl peptidase IV inhibitors (Shoombuatong et al., 2015a) and metabolic syndrome (Worachartcheewan et al., 2013). This study employs Weka’s (Hall et al., 2009) J48 algorithm (a Java implementation of the C4.5 algorithm) for constructing a predictive model for discriminating influenza virus type A and B into its class (active or inactive group). The model is constructed as a function of a set of thirteen molecular descriptors. In the J48 algorithm, the information gain is used to rank features for constructing a decision tree based on feature usage. The feature usage score can be obtained after constructing a decision tree and then counting the firing frequency of associated rules (nodes). The feature usage provides an easy way to rank and identify important features. A molecular descriptor with a high feature usage is considered to be an important feature.

Performance of the J48 algorithm was benchmarked against other commonly used classifiers such as the Naive Bayes (Bayes), support vector machine (SVM) and artificial neural network (ANN). The first classifier that was employed is SVM, which is a statistical learning method that had demonstrated wide utility in QSAR/QSPR modeling. A non-linear SVM was used herein by applying the radial basis function kernel to transform the original feature space to a higher dimensional space in which the SVM classifier linearly separate the inherent classes of the dependent variable via a maximum margin separating hyperplane. A grid search for the optimal parameters was performed according to a 10-fold cross-validation (10-fold CV) scheme using the R package e1071 (Meyer et al., 2008). The second classifier employed herein is ANN, which is based on a back-propagation implementation of the feed-forward neural network approach. ANN had also received widespread popularity in QSAR/QSPR modeling. The suitable number of hidden nodes of the ANN classifier was decided based on the best evaluated 10-fold CV. The Bayes classifier is a statistical classifier that can predict the bioactivity of interest using the probability of class membership according to the Baye’s Theorem. ANN and Bayes classifiers were calculated with default parameters using Weka, version 3.6.12.

Statistical assessment

Four measurements were used to evaluate the prediction performance of the proposed model namely accuracy (Ac), sensitivity (Sn), specificity (Sp) and Matthews correlation coefficient (MCC), which are defined by the following equations: (2) Ac=TP+TNTP+TN+FP+FN

(3) Sn=TPTP+FN

(4) Sp=TNTN+FP

(5) MCC=TP×TN−FP×FNTP+FPTP+FNTN+FPTN+FN

where TP, TN, FP and FN are the numbers of true positive, true negative, false positive and false negative, respectively.

Maximum common substructure analysis

The chemical substructure analysis or molecular fragment analysis was performed to analyze the properties of the NAIs expressed by molecular descriptors using LibMCS software as implemented in ChemAxon’s JChem technology to identify and display the maximum common substructures of compounds in the data set (version 15.1.12.0; ChemAxon Kft., Budapest, Hungary). In brief, all chemical structures in SMILES format were initially converted to SDF format as an input file using MolConverter (version 15.1.12.0; ChemAxon Kft., Budapest, Hungary). LibMCS subsequently generated maximum common substructures present in the data set. The fragments were ranked according to structure-based hierarchical clustering algorithms, in which the bottoms of the hierarchy are the initial structure and then the next level contains the maximum common substructures of initial molecule clusters where all molecules that share the same common structure are placed in a cluster. Active and inactive fragments were distinguished according to pIC50 cut-off values of >6 and <5, respectively and the chemical substructures were ranked according to their fragment occurrence in both the active and inactive groups of the data set.

Binding analysis

To further understand the protein-ligand interaction site, a structure-based molecular docking approach was employed in this study. Sets of 148 and 45 active NAIs against influenza A and B, respectively, were subjected to docking with neuraminidase. In this study, crystal structures of 2009 pandemic H1N1 neuraminidase for influenza A (PDB accession code 3TI4) and B (PDB accession code 1A4G) were retrieved from the Protein Data Bank (Berman et al., 2000). The proteins were initially prepared by removing water molecules and alternative side chains. Hydrogens and Gasteiger charges were added to the protein, which were subsequently cleaned up by merging the charges, repairing bonds and removing non-polar hydrogens and lone pair atoms. Low-energy conformers of active NAIs obtained from the geometrical optimization process were employed to dock with the binding site of neuraminidase. Grid boxes with dimensions of 40 × 30 × 32 Å and 40 × 40 × 40 Å was applied to center the ligands inside the active sites of neuraminidase for influenza A and B, respectively. Molecular docking was performed using AutoDock Vina (Trott & Olson, 2010) with default parameters. Prior to performing docking of the active NAIs, the docking protocols were initially validated by calculating the root-mean-square deviation (RMSD) of atomic positions between co-crystallized ligand and re-binding ligand, which are laninamivir octanoate and zanamivir for PDB accession codes 3TI4 and 1A4G, respectively. The protocol was acceptable with an RMSD value ≤2.0 Å, which was observed to be 1.153 and 1.277 Å for 3TI4 and 1A4G, respectively. Binding energy (Kcal/mol) of ligand conformers were calculated and the conformation providing the lowest binding energy was chosen for further analysis of the binding modality in the active site of neuraminidase. All protein structures were visualized and rendered in PyMOL, version 1.7.6.3.

Combinatorial library enumeration

Novel NAIs were generated via combinatorial library enumeration using AutoGrow 3.0 (Durrant, Amaro & McCammon, 2009; Durrant, Lindert & McCammon, 2013) in which compounds are enumerated inside the binding pocket of influenza neuraminidase. AutoGrow is an evolutionary algorithm that generate populations of ligands through three operators including mutation, crossover and selection. Firstly, the mutation process derives a novel compound by randomly replacing certain moieties with chemically synthesizable click-chemistry reactive groups. Secondly, crossover operator is applied via the alignment of two ligands in order to identify the maximum common substructure, which will serve as the core scaffold by which fragments will be attached to. Newly generated ligands are subsequently filtered following the Lipinski’s rule of five where a relaxed criteria was applied so that the generated compounds could violate not more than one rule of the rule of five. Subsequently, compounds are subjected to a second filter using the Ghose criteria (Ghose, Viswanadhan & Wendoloski, 1999) in which enumerated compounds possessing LogP in the range of −0.4–5.6 were acceptable. Thirdly, a set of ligands obtained from the mutation and crossover operators are then evaluated by the binding fitness against the active site of neuraminidase via selection operators in AutoDock Vina. The top ten lowest binding energy was selected to serve as founders of the next generation. These procedures are repeated in an iterative manner until the specified number of generation has been reached.

Molecular structures of active compounds against both types of influenza neuraminidases were used as molecular founders in the enumeration of a new population of ligands. Grid box was defined to provide coverage of the active site of neuraminidase as well as loop positions 150 and 430 (i.e., critical for enzyme-substrate recognition and may possibly improve the binding fitness of enumerated ligands) (Amaro et al., 2011). The enumeration was set to generate 10 mutants and 10 crossovers per generation such that the top ten ligands with the best AutoDock Vina docking score were advanced to the next generation. Fragment addition were selected from AutoGrow 3.0 fragment library with molecular weight of less than 150 Da. Finally, a total number of 19 and 31 candidate ligands against neuraminidase of influenza A and B, respectively, were enumerated and their binding modalities were subsequently analyzed. Moreover, post-filtering of enumerated ligands against influenza A and B neuraminidase were separately performed using J48 models of influenza A and B NAIs. Molecular structures of the filtered enumerated ligands against influenza A and B neuraminidase are provided as supplementary data in SDF file format on figshare and is accessible at http://dx.doi.org/10.6084/m9.figshare.1612484.

Results and Discussion

Exploratory data analysis

A total number of 313 NAIs were collected from the BindingDB that consisted of 285 and 131 NAIs against influenza A and B, respectively, as shown in Table 1. Because NAIs are used to inhibit neuraminidase from both influenza A and B, however their distinct protein structures give rise to different efficacy of treatment, therefore the two neuraminidases were analyzed separately as to gain a better understanding of their individual pharmacokinetic properties. To determine the different characteristics between active and inactive NAIs, an exploratory data analysis of the thirteen descriptors was carried out via statistical analysis. Summary of the statistical parameters of the underlying data are shown in Table 2 and Tables S1 and S2. Furthermore, histogram plots showing the general data spread of the physicochemical descriptors are provided in Figs. 2 and 3. The p-value was used to compare the 13 descriptors in active and inactive groups of influenza A and B NAIs. The differences between active and inactive groups were considered statistically significant at p ≤ 0.05. Results from the t-test along with their p values are shown in Table 2 for NAIs against influenza A and B.

Table 2 Summary of statistical analysis of active and inactive classes of influenza A and B neuraminidase inhibitors.

Descriptor	Influenza A	Influenza B	
	Active	Inactive	p-value	Active	Inactive	p-value	
MW	343.234 ± 56.916	328.415 ± 83.823	0.085	312.466 ± 44.641	357.692 ± 76.866	<0.05	
RBN	7.061 ± 2.183	5.248 ± 2.681	<0.05	5.689 ± 2.043	7.233 ± 2.561	<0.05	
nCIC	1.514 ± 0.655	2.109 ± 1.316	<0.05	1.444 ± 0.546	1.709 ± 0.852	<0.05	
nHDon	4.743 ± 1.257	4.321 ± 2.029	<0.05	4.578 ± 1.34	4.849 ± 1.561	0.302	
nHAcc	7.777 ± 1.502	7.204 ± 2.153	<0.05	7.156 ± 1.731	8.337 ± 1.334	<0.05	
ALogP	0.049 ± 1.256	0.853 ± 2.475	<0.05	−0.128 ± 1.291	−0.017 ± 1.602	0.67	
TPSA	125.205 ± 24.37	120.033 ± 37.06	0.169	114.592 ± 27.661	134.168 ± 25.007	<0.05	
Qm	0.163 ± 0.02	0.179 ± 0.055	<0.05	0.158 ± 0.013	0.168 ± 0.018	<0.05	
Energy	−0.274 ± 0.083	−0.246 ± 0.119	<0.05	−0.276 ± 0.099	−0.249 ± 0.089	0.129	
Dipole moment	4.101 ± 1.972	4.328 ± 2.003	0.336	3.831 ± 1.601	4.364 ± 2.011	0.101	
HOMO	−0.352 ± 0.011	−0.343 ± 0.015	<0.05	−0.351 ± 0.009	−0.35 ± 0.013	0.573	
LUMO	−0.006 ± 0.018	−0.021 ± 0.022	<0.05	−0.013 ± 0.01	−0.003 ± 0.023	<0.05	
HOMO-LUMO	0.346 ± 0.023	0.322 ± 0.03	<0.05	0.339 ± 0.014	0.347 ± 0.029	<0.05	

Figure 2 Histogram representing the molecular descriptors for NAIs against influenza A.

Note: Active and inactive NAIs are represented with red and blue bars, respectively, whereas their overlapping region are shown in purple.

Figure 3 Histogram representing the molecular descriptors for NAIs against influenza B.

Note: Active and inactive NAIs are represented with red and blue bars, respectively, whereas the purple represents their overlap region.

Exploratory data analysis showed that the NAIs displayed drug-like properties according to Lipinski’s rule of 5 (Lipinski et al., 2001) in which compounds generally exhibit the following features: (1) MW <500 Da, (2) LogP <5, (3) nHDon <5 and (4) nHAcc <10. An in-depth analysis of the molecular descriptors as a function of active and inactive NAIs were carried out as to shed light on the origin of the neuraminidase inhibitory activity. As described in the Materials and Methods, compounds were classified as active or inactive using pIC50 cut-offs of ≥ 6 (IC50≤1μM) and ≤ 5 (IC50 ≥10μM), respectively; however, compounds that exhibited a pIC50 value in the range of 5–6 were not considered in this study (similar to the ‘Data collection’). The bioactivities of the NAIs were determined by observing the mean pIC50 value, which was 5.788 ± 2.023 (1.30 µM) and 5.107 ± 1.695 (7.80μM) for type A and B neuraminidase, respectively. It could be observed that NAIs for influenza A neuraminidase possessed significantly different therapeutic activity than those for type B neuraminidase with p < 0.05.

MW refers to the molecular size of compounds and is an important parameter of Lipinski’s rule of five for drug-like molecules. Statistical analysis showed that the average molecular size of active compounds for influenza A NAIs (343.234 ± 56.916) was not significantly different from that of inactive compounds (328.415 ± 83.823) with p = 0.085. However, for influenza B NAIs, the average MW of the active (312.466 ± 44.641) and inactive (357.692 ± 76.866) groups were significantly different with p < 0.05.

RBN is the number of rotatable bonds in a molecule and provides a relative measure of molecular flexibility. RBN is defined as any single bond, not in a ring, that is bound to a non-terminal heavy atom. Amide C–N bonds are excluded from the count because of their high rotational energy barrier. As shown in Table 2, the number of rotatable bonds in a molecule of the active group (7.061 ± 2.183) for influenza A NAIs is notably different from that of the inactive group (5.248 ± 2.681). In the case of influenza B NAIs, the active group (5.689 ± 2.043) is also different from the inactive group (7.233 ± 2.561) as shown in Table 2.

nCIC is calculated as the cardinality of the set of independent rings known as the smallest set of smallest rings. As shown in Table 2 and Table S1, the average number of rings of the active group (1.514 ± 0.655) of influenza A NAIs is less than that of the inactive group (2.109 ± 1.316). Similar to type B, the average number of rings of the active group (1.444 ± 0.546) is not greater than that of the inactive group (1.709 ± 0.852) at p < 0.05.

nHDon refers to the number of hydrogen bond donors in a molecule. In brief, the active group was found to possess higher mean values of nHDon than the inactive group for influenza A NAIs, where as for influenza B NAIs, the active group was found to possess lower mean values of nHDon than the inactive group. As shown in Fig. 2, the histograms of nHDon in the active/inactive groups indicate that the distributions for influenza A NAIs are significantly different, whereas the distributions for influenza B NAIs are not significantly different at p < 0.05.

nHAcc describes the number of hydrogen bond acceptors in a molecule. Table 2 shows that nHAcc of the active group for influenza A NAIs (7.777 ± 1.502) is greater than that of the inactive group (7.204 ± 2.153). Similar to influenza B NAIs, the numbers of nHAcc of the active (7.156 ± 1.731) and inactive groups (8.337 ± 1.334) were statistically different (at p < 0.05) as also indicated by histogram plots in Figs. 2 and 3 for influenza A and B, respectively.

ALogP is a computational method for estimating the 1-octanol/water partition coefficient (logP), which is a well-known measure of the molecular hydrophobicity also known as lipophilicity. As shown in Figs. 2 and 3, the histogram of ALogP of influenza B has a greater overlapping region (purple) than that of type A. Table 2 shows that ALogP was statistically significant for actives versus inactives for NAIs of both influenza. Mean AlogP values of 0.049 ± 1.256 and 0.853 ± 2.475 were observed for active and inactive influenza A NAIs, respectively, whereas values of −0.128 ± 1.291 and −0.017 ± 1.602 were observed for influenza B NAIs, respectively.

TPSA describes the contribution of polar atoms to the molecular charge based on an empirical measurement of the polar surface area of a molecule. Table 2 shows that actives and inactives for influenza type A NAIs is not statistically significant at the p < 0.05 level while the active and inactive groups of influenza B NAIs were statistically significant. The corresponding TPSA values were 125.205 ± 24.370 (active) and 120.033 ± 37.060 (inactive) for influenza A NAIs, whereas the TPSA values of influenza B NAIs were 114/592 ± 27.661 (active) and 134.168 ± 25.007 (inactive).

Mean absolute charge (Qm) describes the global measurement of the molecular charge. The histogram plot showed different distributions of Qm for influenza A and type B NAIs. Moreover, Qm exhibited a distinct Qm of 0.171 ± 0.042 and 0.165 ± 0.017 for influenza A and type B NAIs, respectively. This study suggested that the inactive group had higher Qm values when compared to the active group as shown in Table 2. The Qm values were statistically significant at p < 0.05 for the active and inactive influenza A NAIs were 0.163 ± 0.020 and 0.179 ± 0.055, respectively, while values of 0.158 ± 0.013 and 0.168 ± 0.018 were observed for the active and inactive groups of influenza B NAIs, respectively.

Energy represents the summation of the atomic energy. Overall, no significant difference in energy were observed for NAIs of influenza A (−0.260 ± 0.103) and B (−0.258 ± 0.093) with p = 0.823. It was found that the active group (−0.274 ± 0.083) had a slightly higher energy than the inactive group (−0.246 ± 0.119) for influenza A. However, the values of (−0.276 ± 0.099) and (−0.249 ± 0.089), which were observed in the active and inactive groups, respectively, of influenza B NAIs were not statistically significant at p < 0.05 (p = 0.129).

Dipole moment represents the asymmetric distribution of charge in a molecule. A high dipole moment value indicates a high charge distribution and vice versa. The dipole moment of NAIs for influenza A (4.210 ± 1.987) and B (4.181 ± 1.891) were not significantly different at p < 0.05 (p = 0.737). Further statistical analysis revealed that the dipole moment of actives versus inactives for both influenza A and B NAIs are also not significantly different, with p = 0.336 and p = 0.537, respectively.

The highest occupied molecular orbital (HOMO) and lowest unoccupied molecular orbital (LUMO) are the highest- and lowest-energy molecular orbitals that are occupied and unoccupied by electrons, respectively. HOMO is associated with ionization potential (ability to donate electrons), whereas LUMO is responsible for electron affinity (ability to accept electrons). It could be observed that the HOMO value of the active group is significantly different from that of the inactive group with p < 0.05. Moreover, in the case of influenza B NAIs, the HOMO value of the active group is not statistically significant (p = 0.573) at p < 0.05 as summarized in Table 2. However, the mean values of LUMO for influenza A and B NAIs were statistically significant when comparing the active and inactive groups at p < 0.05 as summarized in Table 2.

HOMO-LUMO describes the kinetic stability and chemical reactivity of molecules. A small energy gap between these two states pertains to low kinetic stability and provides high chemical reactivity and vice versa (Mathammal et al., 2015). The histogram plot shows a slightly different pattern of distribution of the NAIs for influenza A and B between the active and inactive groups (Figs. 2 and 3, respectively). The overviews of the HOMO-LUMO values of influenza A (4.210 ± 1.987) and B (4.181 ± 1.891) NAIs were significantly different at the p = 0.737 level. For analysis of influenza A and B NAIs, the HOMO-LUMO value of the active group is shown with the statistically significant results at the p < 0.05 level for both type A and B as summarized in Table 2.

In summary, these results indicated that several descriptors were significant for differentiating the active and inactive class of influenza A NAIs (p < 0.05) except for MW, TPSA and dipole moment. Similarly, several descriptors were also significant for differentiating the active and inactive class of influenza B NAIs with the exception of nHDon, ALogP, energy, dipole moment and HOMO descriptors. The active class of influenza A NAIs tended to have higher values of flexibility (RBN), hydrogen-bond donors (nHDon) and acceptors (nHAcc) as well as chemical reactivity/stability (HOMO-LUMO) while possessing low values for lipophilicity (ALogP), ring moiety (nCIC), molecular energy and charge (Qm). On the other hand, the active class of influenza B NAIs had the tendency of being smaller in size (MW), less molecular flexibility (RBN), ring moiety (nCIC), hydrogen-bond acceptors (nHAcc), charge (Qm) but higher chemical reactivity/stability of molecules (HOMO-LUMO). Nevertheless, in practice, compounds used for treating influenza B are the same compounds used to develop treatments for influenza A. As the univariate analysis can only provide an overview of the general features of NAIs therefore multivariate analysis was performed to construct predictive models using substructure descriptors for classifying the bioactivity of NAIs as well as discerning key features for differentiating active NAIs from their inactive counterparts.

PCA analysis

Furthermore, PCA was applied to explore the chemical space of NAIs for influenza A (Figs. 4A and 4B correspond to scores and loadings plots, respectively) and B (Figs. 4C and 4D correspond to scores and loadings plots, respectively) whereby the set of 13 descriptors were mapped onto a few PCs. PC1 had the highest variance in the data of influenza A NAIs with a value of 38.59%. Meanwhile, PC2 and PC3 provided the second and third highest variances with values of 19.87% and 12.25%, respectively. In summary, the first three PCs afforded a cumulative variance of 70.71%, which was sufficiently informative for further analysis. Particularly, the loadings plot (Fig. 4B) shows that ALogP, HOMO and nCIC dominated the periphery of the negative end of PC1 while TPSA, nHDon, nHAcc and HOMO-LUMO were distributed on the positive end thereby suggesting the importance of these descriptors in accounting for the variance of PC1. Furthermore, molecular energy and HOMO-LUMO dominated the negative end while MW was found towards the terminal side of the positive end of PC2. Moreover, the variance of PC3 was accounted for by Qm on the negative terminal while the LUMO on the positive periphery. Taken together, the cluster of descriptors comprising of nHAcc, nHDon and TPSA together with the cluster of descriptors consisting of RBN, LUMO and HOMO-LUMO (i.e., both shown as green clusters) were found to characterize the features of active compounds owing to its distribution in the spatial location of the active compounds when the scores and loadings plots are superimposed. It can thus be seen that active compounds were accounted by hydrogen bond propensities, molecular orbital energies as well as the rotatable bond count and polar surface area. Likewise, the other cluster of descriptors consisting of ALogP, HOMO, nCIC and molecular energy were found to define the inactive set of compounds as indicated by the red cluster, which corresponded to lipophilicity, electron donating propensity and number of rings.

Figure 4 PCA scores and loadings plots of NAIs against influenza A (A and B, respectively) and B (C and D, respectively).

Active and inactive compounds are represented by green and red circles, respectively, in the scores plots. Important features for rationalizing the active and inactive compounds are highlighted by green and red clusters, respectively. An interactive version is available at https://dx.doi.org/10.6084/m9.figshare.3123136.v1.

As for influenza B NAIs, PC1 afforded the highest variance with a value of 37.66%. Meanwhile, PC2 and PC3 provided the second and third highest variances with values of 23.02% and 9.88%, respectively. It was found that the first three PCs accounted for 70.56% of the total variance. Particularly, the positive end of PC1 was dominated by TPSA, nHDon, nHAcc, HOMO-LUMO and LUMO, respectively, while the negative end by HOMO. Furthermore, descriptors at the periphery of the positive end of PC2 were comprised of MW, nCIC, ALogP and RBN while having Qm dominating the negative end. Moreover, dipole moment and molecular energy were found as terminal descriptors at the positive end of PC3 with RBN and MW on the negative end. Putting these findings into perspective, it can be seen that the first PC was accounted by molecular orbital energies and hydrogen bond donating and accepting propensities, the second PC was characterized by structural features comprising of the molecular weight, number of rings, lipophilicity and rotatable bond count while similarly the third PC was rationalized by the rotatable bond count and molecular weight. The loadings plot (Fig. 4D) shows two distinctive cluster of descriptors as indicated by the green and red clusters, which were found to characterize compounds as being active and inactive, respectively. Comparison of the loadings plots of influenza A and B NAIs revealed that TPSA, nHAcc, nHDon, HOMO-LUMO and LUMO were important features for characterizing active compounds. Likewise, four features comprising of ALogP, HOMO, nCIC and molecular energy were crucial for discriminating compounds as inactive.

Table 3 Summary of performance comparison of decision tree algorithm with other learning methods for classifying the bioactivity of influenza A and B NAIs.

Classifier	Data subset	Influenza A	Influenza B	
		Ac (%)	Sn (%)	Sp (%)	MCC	Ac (%)	Sn (%)	Sp (%)	MCC	
Bayes	Training set	86.84	92.37	80.91	0.74	98.10	100.00	97.10	0.96	
	10-fold CV set	81.58	85.59	77.27	0.63	98.10	100.00	97.10	0.96	
	External set	78.95	100.00	55.56	0.63	38.46	100.00	5.88	0.15	
ANN	Training set	98.68	99.15	98.18	0.97	98.10	97.22	98.55	0.96	
	10-fold CV set	88.16	90.68	85.45	0.76	83.81	75.00	88.41	0.64	
	External set	91.23	86.67	96.30	0.83	96.15	88.89	100.00	0.92	
SVM	Training set	98.68	99.15	98.18	0.97	98.10	97.22	98.55	0.96	
	10-fold CV set	89.04	94.92	82.73	0.78	84.76	61.11	97.15	0.66	
	External set	92.73	100.00	84.00	0.86	88.46	66.67	100.00	0.75	
DT	Training set	92.98	96.61	89.09	0.86	96.19	97.22	95.65	0.92	
	10-fold CV set	88.60	91.53	85.45	0.77	81.90	61.11	92.75	0.58	
	External set	89.47	83.33	96.30	0.80	96.15	100.00	94.12	0.92	

Prediction of inhibitory activity against neuraminidase from influenza A and B

An interpretable predictive model is more useful for providing insights into the basis of the biological and chemical properties of influenza A and B NAIs. Therefore, in this study, a QSAR model based on the J48 algorithm is presented for discriminating between active/inactive groups of influenza A and B NAIs. Each compound was calculated as an M-dimensional vector of 307 bits based on substructure fingerprints. In constructing the predictive model, the J48 algorithm was applied using the encoded compounds from the internal sets. Moreover, to evaluate the ability of our proposed QSAR model, two different experiments were performed: one experiment was performed on the full training data and the other experiment was evaluated using a 10-fold CV procedure as shown in Table 3. The CV procedure was performed by first partitioning the data into 10 equally sized segments or folds; then, 9 folds were used as the training data while the remaining fold was used for validation. Finally, the results were averaged across the 10 experiments. Four measurements were used to assess the performance of the QSAR models namely Ac, Sn, Sp and MCC.

Table 3 demonstrated that using the set of 307 descriptors provided promising results with an Ac of 88.60%, Sn of 91.53%, Sp of 85.45% and MCC value of 0.77 for influenza A NAIs, whereas these descriptor set also performed well for influenza B NAIs with an Ac of 81.90%, Sn of 61.11%, Sp of 92.75% and MCC value of 0.58. As shown in Table 1, the used data set is not balanced because the number of positive samples (active group) is larger than that of negative samples (inactive group). Therefore, the Sn is considerably greater than the Sp for influenza A NAIs. To address this problem, the original data set should first be balanced between the active and inactive groups. In addition, to assess the reliability of the predictive model on unknown data, an external set was considered. Table 3 shows that our proposed model still performs well for predicting influenza A NAIs with an Ac of 89.47%, Sn of 83.33%, Sp of 96.30% and MCC of 0.80 while the performance for predicting influenza B NAIs was acceptable with an Ac of 96.15%, Sn of 100.00%, Sp of 94.12% and MCC of 0.92. It was well recognized that a decision tree-based classifier utilized the estimated threshold to predict a sample. Moreover, the employed data set is not balanced in which the number of positive samples (active) is smaller than that of negative samples (inactive). Thus, it was not surprising that the prediction result of our proposed model on type B provided a moderate Sn of 61.11%. However, our proposed model aims to maximize both the simplicity and interpretability of the classification method. The classification tree of NAIs against both type of influenza neuraminidase were illustrated in Fig. 5.

Figure 5 Illustration of decision tree model for classifying the activity of NAIs against Influenza types A and B as a function of their substructure fingerprint.

The full descriptive name of the substructure fingerprints are shown for the purpose of clarity while their corresponding acronyms are provided in the text as well as the supplementary data available on figshare at http://dx.doi.org/10.6084/m9.figshare.1612484. It should be noted that “1,3-tautomerizable” and “chiral center specified” correspond to idiosyncratic PaDEL definitions rather than “standard definitions”.

The J48 algorithm was benchmarked against other commonly used learning methods such as Bayes, SVM and ANN classifiers. For fair comparisons, the other classifiers were constructed using the same set of thirteen descriptors. Table 3 shows the comparative results on the influenza A and B NAIs data sets. Predictive performance for the external set of influenza A for Bayes, SVM, ANN and J48 classifiers had Ac values of 78.95%, 92.73%, 91.23% and 89.47%, respectively, while affording MCC values of 0.63, 0.86, 0.83 and 0.80, respectively. Meanwhile, the influenza B data set afforded Ac values of 38.46%, 88.46%, 96.15% and 96.15%, respectively, while MCC values were 0.15, 0.75, 0.92 and 0.92, respectively. Comparisons of the performance can be briefly summarized as follows. In predicting the bioactivity of influenza A, the SVM classifier performed well with the highest external Ac while the J48 classifier was comparable with such model. As for the influenza B data set, the highest external MCC was achieved by the J48 and SVM classifiers. The ANN classifier performed well with the second highest MCC. The aforementioned results revealed that the J48 algorithm was comparable to that of the SVM classifier on the influenza A data set and also outperformed other benchmarked classifiers.

Substructure fingerprint play an important role in representing the characteristics of compounds. Thus, the identification of informative substructure fingerprints would help provide insights into the underlying mechanism of influenza A and B NAIs. The feature importance plot is shown in Fig. 6 where features with the largest descriptor usage are deemed to be the most important. Figure 6A shows that the top three informative fingerprints of influenza A NAIs are SubFPC100, SubFPC41 and SubFPC300, which corresponds to secondary amide, 1,2-Diol and 1,3-tautomerizable moiety, respectively. Moreover, Fig. 6B shows that the top three informative fingerprints of influenza B NAIs are SubFPC5, SubFPC14 and SubFPC88, which corresponds to alkene, secondary alcohol and carboxylic acid derivative moiety, respectively. Secondary amide was found as the root node of the decision tree for classifying the bioactivity of influenza A NAIs followed by the 1,2-diol moiety (Fig. 5A). Notably, compounds lacking secondary amide were classified as inactives whereas compounds possessing the 1,2-diol moiety were classified as inactive NAIs. On the other hand, the alkene moiety can be considered to be an important substructure for classifying the bioactivity of influenza B NAIs as it was found to be the root node of the decision tree. Nevertheless, it can be seen that the decision rule for classifying the bioactivity of influenza B NAIs are reliant on several substructure fingerprints, which is more than that of influenza A NAIs (Fig. 5B).

Figure 6 Plots of the descriptor usage derived from the decision tree model.

Descriptors with the largest percentage of descriptor usage is deemed the most important.

Analysis of maximum common substructure

The molecular substructure analysis revealed the important molecular fragments that facilitate the biological activity against influenza neuraminidase. The top-ranking substructures for both active and inactive NAIs agianst influenza A are indicated in Figs. 7A and 7B, respectively, while Fig. 7C and 7D represents active and inactive NAIs against influenza B, respectively. The top five substructures were sorted by substructure occurrence. The results of the top five active fragments indicated that cyclohexene-based, dihydropyran-based and cyclopentane-based scaffolds are relevant to inhibitory activity against influenza neuraminidase, in which these six- and five-membered non-aromatic rings possess a marginal ligand-binding conformation comparable to the tetrahydropyran ring of the sialic acid substrate of influenza neuraminidase.

Figure 7 Summary of common substructure in active and inactive sets of NAIs against influenza A (A and B, respectively) and B (C and D, respectively).

Number of substructure occurrences are indicated in bracket below the substructure’s rank.

The top-ranked common substructure was a cyclohexene-based moiety, which can be found in the current drug of choice for influenza treatment: oseltamivir. This drug was developed to lower the polarity effect of the dihydropyran scaffold of the first-generation NAIs, which led to the low bioavailability as observed in zanamivir. Initially, zanamivir was developed based on a dihydropyran scaffold and exerts good inhibitory activity against influenza neuraminidase (Meindl et al., 1974; Von Itzstein et al., 1993), which became the first approved NAI for use as a therapeutic agent against the influenza virus. Structure-based drug design based on the availability of N2 sialidase X-ray co-crystal structure with α-Neu5Ac and Neu5Ac2en (Varghese et al., 1992) was used as guidelines for the development of novel NAIs. In silico analysis of enzyme active sites revealed energetically favorable interactions of amino acid residues in the active site and various functional group probes, such as carboxylates, amines, methyl groups and phosphates (Von Itzstein et al., 1996). The molecular structure overlay of predicted favorable functional groups against co-crystal structure of N2 sialidase and Neu5Ac2en as template molecules suggested that substituting the C-4 hydroxyl group of the template with amino and guanidino groups should improve the binding affinity with the N2 active site. As a result of amino substitution at the C4 hydroxyl group, the binding affinity is enhanced by the formation of a salt bridge between the amino group and E199 residue, whereas guanidino substitution interacts with E119 and E227 via its terminal nitrogen (Von Itzstein et al., 1993; Von Itzstein et al., 1996). Nevertheless, this acid–based inhibitor processed high polarity due to the ring oxygen and polar glycerol side chain, resulting in low bioavailability. Thus, this drug was considered to be administered by inhalation, which is difficult to provide in some patients, particularly children. The development of orally administrated NAIs was required to overcome this problem.

As previously mentioned, the polarity of dihydropyran-based NAIs affects their pharmacokinetic properties and the route of administration. To reduce the polarity effect of the dihydropyran scaffold, scaffold hopping was employed to identify appropriate molecular scaffolds that would exert desirable properties. A cyclohexene scaffold was used to replaced the ring oxygen, which was previously reported to be a non-essential moiety required for neuraminidase inhibition (Taylor & Von Itzstein, 1994). Replacing dihydropyran with a cyclohexene ring in which the double bond position is similar to the sialosyl transition state provided significantly higher inhibitory activity (Kim et al., 1997). Moreover, the glycerol side chain is also considered to be a main source of polarity due to its high number of oxygen atoms. The modification of the hydrophilic glycerol side chain with a 3-pentyl ether side chain based on the structure–activity relationship study led to the development of GS 4071, which was subsequently named oseltamivir carboxylate, a potent sialidase inhibitor. As a result of introducing the 3-pentyl ether side chain, the binding interaction is reorganized by reorientation of E276 from this side chain to form a salt bridge with R224, leading to the generation of a substantial hydrophobic patch, which increases the binding affinity with the ligand’s hydrophobic side chain (Itzstein & Thomson, 2009). Elimination of the oxygen atom in combination with functional group modification led to lower polarity and increased the bioavailability of molecules. Thus, the second NAI was developed and consequently approved, named oseltamivir, which is currently used as a drug of choice for treating influenza. In addition, the successful development of cyclohexene-based NAIs results in the generation of extensive studies for developing novel NAIs using this scaffold.

Recently, the cyclopentane scaffold in furanose was found to possess an inhibitory effect against influenza neuraminidase as strongly as the lead compound of sialidase inhibitor, called DANA. The report on inhibitory activity by furanose revealed the potential of cyclopentane as a novel scaffold for the development of NAIs (Yamamoto et al., 1992). Structure-based analysis of the cyclopentane scaffold using protein crystal structure information indicates a distinct binding mode, in which the cyclopentane ring re-organized the functional groups of NAI to interact with amino acid residues inside the binding pocket of influenza neuraminidase (Stoll et al., 2003). This evidence revealed an opportunity for introducing NAIs with novel scaffolds. The most recently approved NAI, named peramivir, was developed based on a five-membered ring scaffold. A set of novel NAIs with five-membered ring scaffolds were synthesized using cyclopentane derivatives incorporating three functional group substitutions of zanamivir, which included carboxylate group, C5-acetamido group and C4-guanidino group, arranged in all expected positions inside the N9 active site. The functional group binding with the negatively charged area in the active site, which previously interacted with the C4 hydroxyl group of Neu5Ac2en, was designed to replace with a guanidino group as similarly observed in zanamivir. The addition of n-butyl was designed to interact with the hydrophobic region, which was previously occupied by the glycerol side chain of Neu5Ac2en. The binding interaction was confirmed by co-crystallization with N9 sialidase and the crystal structure indicates that the binding interactions are comparable with those of zanamivir (Babu et al., 2000).

Nevertheless, some of the molecular substructures that were present in the active group of NAIs, such as 3-acetamido-2-methyl-3,4-dihydro-2H-pyran-6-carboxylic acid and 5-amino-4-acetamidocyclohex-1-ene-1-carboxylic acid, can be found in the inactive group of influenza A and B neuraminidase, respectively. Note that the inhibitory activities against influenza neuraminidase are facilitated by additional factors from both protein and ligand sides. From the protein perspective, the neuraminidase share approximately 90% structural homology in the same subtype, whereas the homology between subtypes is lower, 50% and 30%, between influenza A and B (Shtyrya, Mochalova & Bovin, 2009). The distinct structural homology affects the conformation of catalytic residues inside the catalytic pocket, resulting in different fitness binding of ligands. On the other hand, the composition of the ligand and properties affect the efficiency of the binding interaction. These factors are frequently observed by the type and position of functional groups lying in molecules, which are the crucial part for interacting with the target enzyme for inhibition. The overall size of the molecules and the molecular conformations are also important for binding with the enzyme because the binding pocket has a unique geometrical conformer that limits the shape and electrostatic properties of the target molecules. In addition, the drug-like properties of the ligand also facilitate pharmacokinetics and pharmacodynamics of ligands to reach their target and generate desirable bioactivity for therapeutic purposes.

Analysis of binding modality

The observations on the active set of NAIs fragments revealed a pattern of molecular scaffolds that exhibited activity against the neuraminidase glycoprotein of influenza A. Note that the molecules shared a similar conformation substructure as the original substrate, sialic acid and tended to exhibit inhibitory potential against this enzyme. The binding pocket in the active site of neuraminidase contains eight highly conserved amino acid residues, which interact with the substrate and provide catalytic activity in the binding pocket. These residues can be grouped into five minor sites as illustrated in Fig. S1. Thus, designing novel NAIs requires choosing functional groups that can interact and fit with these sites of conserved residues to prevent catalytic reactions with this enzyme. To investigate the binding modes of active compounds against neuraminidase of both influenza A and B, a combination of molecular docking and post-docking analysis using AutoDock Vina (Trott & Olson, 2010) and SiMMAP web-server (Chen et al., 2010), respectively, was employed to identify key interactions and important moieties facilitating protein-ligand interactions.

Figure 8 Binding modes of NAIs in active site of influenza A and B neuraminidase are shown in (A) and (B), respectively.

Electrostatic (Elec), hydrogen-bond (Hbond) and van der Waals’ (vdW) interaction sites are indicated by red, blue and orange sphere, respectively. Interacting residues (N2 numbering) of Elec, Hbond and vdW are highlighted in white, cyan and yellow, respectively.

The analysis of 148 active NAIs against influenza A revealed four distinct binding anchors (Elec1, vdW1, vdW2 and vdW3) with their site-moiety preferences as illustrated in Fig. 8A. Elec1 is the first anchor site and it facilitates electrostatic interactions with carboxylic and alkyl phosphate groups of NAIs through the positive charge of the arginine side chain. The amino acid members of this anchor include R118, R292 and R371, which belongs to the S1 subsite of the influenza A neuraminidase active site (Von Itzstein, 2011; Stoll et al., 2003). In contrast, another three anchor sites are facilitated by van der Waals interactions. The first anchor, vdW1, consisted of R152, I222 and E227, which forms the S3 subsite of the neuraminidase binding pocket (Von Itzstein, 2011; Stoll et al., 2003). The moiety preferences of this anchor are composed of a heterocyclic ring, aromatic moiety, phenol group and aliphatic moiety with alkene. Another van der Waals interaction site was found at the vdW2 anchor site, which contains R224, E227 and R292 as key residues. This anchor facilitates van der Waals contact against aliphatic moieties with an alkene linkage, heterocyclic and aromatic moieties. vdW3 is the final anchor site, with a moiety preference of heterocyclic moiety, alkene linkage of aliphatic moiety and formamidine group. These findings have shown that NAIs interact with both functional residues that facilitate enzymatic reactions and structural residues that maintain the active site architecture (Shtyrya, Mochalova & Bovin, 2009).

The analysis of the binding anchor of 45 active NAIs targeting influenza B revealed four different anchor sites of the binding pocket: Elec1, Hbond1, vdW1 and vdW2 as shown in Fig. 8B. Electrostatic interactions between NAIs and amino acid residues primarily occurred with R115, R291 and R373 (comparable to R118, R292 and R371 in N2 numbering), which are members of the Elec1 anchor. The positive charge of the arginine side chain prefers carboxylic groups as its moiety preference. Note that this finding is similar to anchor Elec1 of influenza A neuraminidase. Interestingly, there are several moiety types of NAIs against influenza B virus that tend to form hydrogen bonds with amino acids in the Hbond1 anchor. The phenolic moiety of D148 and the carboxylic side chain of Y408 (comparable to D151 and Y406 in N2 numbering) facilitate hydrogen bonding through amino groups, carboxylic moieties, primary and secondary alcohols and ester moieties. It can be observed that these residues are members of the S2 subsite of influenza neuraminidase and are responsible for catalytic residues essential for enzyme functioning (Shtyrya, Mochalova & Bovin, 2009). Furthermore, van der Waals contact sites are observed at two anchor site: vdW1 and vdW2. The first van der Waals interaction site is facilitated by R149, W176 and R222 (comparable to R152, W178 and R224 in N2 numbering) and their moiety preference is aliphatic moiety with alkene linkage, heterocyclic ring and aromatic moiety. The second van der Waals anchor is facilitated by I220, R222 and E274 (comparable to I222, R224 and E276 in N2 numbering), which have a heterocyclic ring and alkene linkage of aliphatic moiety as their moiety preference. The results of the post-docking analysis revealed the important amino acid residues and their moiety preferences that can generate potential protein-inhibitor complexes to inhibit enzymatic functioning of influenza neuraminidase.

Linking the results from molecular docking and decision tree revealed pertinent knowledge for targeting key residues of neuraminidases from influenza A and B. As mentioned above, the top three informative descriptors of influenza A NAIs were LUMO, nHDon and nHAcc while nHAcc, energy and HOMO-LUMO were informative descriptors for influenza B NAIs. Analysis of the catalytic pocket of neuraminidase from influenza A and B revealed the presence of the following key residues: arginines, aspartic acid, glutamic acids and tyrosine. The neuraminidase from influenza A had five arginines (e.g., R118, R152, R224, R292, R371), one aspartic acid (e.g., D151), two glutamic acids (e.g., E227 and E277) and one tyrosine (e.g., Y406) while five arginines (e.g., R114, R149, R222, R291 and R373), one aspartic acid (e.g., D148), two glutamic acids (e.g., E225 and E274) and one tyrosine (e.g., Y408). Arginines can act as both electron acceptor and hydrogen bond donor and this is well supported by the decision tree model in which LUMO and nHDon were the highest and second highest feature usage for influenza A NAIs. Aspartic and glutamic acids can act as hydrogen bond acceptor, which is supported by the decision tree model where nHAcc were the third highest feature usage for influenza A NAIs and the highest feature usage for influenza B NAIs. Tyrosine can act as both hydrogen bond donor and acceptor, which is in line with nHDon and nHAcc descriptors of influenza A NAIs where it afforded the second and third highest feature usage while the nHAcc descriptor had the highest feature usage for influenza B NAIs.

Enumeration of neuraminidase inhibitors

Enumerating compound library with robust bioavailability is a challenging task in the drug discovery process of NAIs. To address this challenge, the Ghose’s criteria was applied for filtering enumerated ligands in which the lipophilicity was set to be in the range of −0.4–5.6 and molecular weight was set to be in the range of 160–480 Da. Ligand enumeration for both types of influenza neuraminidase was subsequently performed via the automatic evolutionary algorithm of AutoGrow 3.0. The enumerated ligands were subjected to post-filter removal via the decision tree model described above. This resulted in total sets of 17 and 19 drug candidates for targeting the active site of neuraminidase from influenza A and B, respectively. These candidates were also subjected to comparison with FDA-approved drugs (e.g., zanamivir, oseltamivir and peramivir) and the long-acting laninamivir (i.e., pending FDA approval; approved in Japan). Pariticularly, the binding energy and molecular properties of enumerated ligands and reference drugs are compared as shown in Table 4. Herein, the top ten enumerated ligands against influenza A and B neuraminidase were categorized according to their molecular scaffolds as summarized in Fig. 9.

Table 4 Summary of binding energy and physicochemical descriptors of top ten enumerated ligands against influenza A and B neuraminidase.

	Ligands	MW	ALogP	nHDon	nHAcc	nCIC	RBN	TPSA	Binding energy (Kcal/mol)	
Influenza A	A1	483.57	2.512	1	10	3	9	131.44	−9.70	
	A2	448.57	2.503	1	9	3	7	111.24	−9.30	
	A3	457.58	1.782	3	9	3	8	128.03	−9.30	
	A4	463.59	1.381	3	10	3	7	137.26	−9.20	
	A5	485.60	2.624	3	11	3	9	150.25	−9.10	
	A6	338.45	0.952	2	7	2	5	95.94	−8.30	
	A7	429.57	1.156	4	8	3	8	121.96	−8.00	
	A8	387.48	−0.038	4	8	3	7	121.96	−8.00	
	A9	413.47	0.723	2	9	3	8	125.37	−7.90	
	A10	382.53	0.061	5	10	2	6	149.97	−7.60	
	Zanamivir	332.36	−3.669	9	11	1	6	200.72	−7.60	
	Oseltamivir	284.40	0.446	4	6	1	6	101.65	−6.80	
	Peramivir	329.48	−1.646	8	7	1	8	150.27	−7.20	
	Laninamivir	347.40	−4.717	9	10	1	8	188.96	−7.90	
Influenza B	B1	475.49	2.323	6	12	2	9	136.90	−8.10	
	B2	477.51	2.426	6	12	2	9	136.90	−8.10	
	B3	465.56	0.017	6	12	2	9	122.71	−7.80	
	B4	421.52	1.235	6	9	2	9	136.90	−7.80	
	B5	463.54	−0.086	6	12	2	9	122.71	−7.70	
	B6	469.46	2.194	2	12	2	9	125.37	−7.60	
	B7	403.53	1.325	6	8	2	9	136.90	−7.50	
	B8	493.61	2.323	5	11	2	8	113.68	−7.50	
	B9	443.47	1.433	4	11	2	8	121.96	−7.50	
	B10	381.40	0.400	4	11	1	7	121.96	−7.40	
	Zanamivir	332.36	−3.669	9	11	1	6	200.72	−7.50	
	Oseltamivir	284.40	0.446	4	6	1	6	101.65	−7.10	
	Peramivir	329.48	−1.646	8	7	1	8	150.27	−7.60	
	Laninamivir	347.40	−4.717	9	10	1	8	188.96	−7.80	

Figure 9 Molecular structures of enumerated ligands against neuraminidase of influenza A (A1–A10) and B (B1–B10) are categorized according to their scaffold types and compared to FDA-approved drugs (e.g., zanamivir, oseltamivir and peramivir) as well as the long-acting laninamivir.

It should be noted that these enumerated ligands passed the decision tree-based post-filter.

According to Table 4, the predicted ALogP of the top ten enumerated ligands are in the range of −0.038–2.512 and −0.086–2.426 for influenza A and B, respectively. It should also be noted that the average ALogP of enumerated ligands exhibit improved values where it is higher than the currently approved drugs as well as those under development. MW of ligands are in range of 338.45–484.59 Da and 381.40–493.61 Da for influenza A and B, respectively, which is also in agreement with the Lipinski’s rule of five for drug-like molecules. Furthermore, almost all of the top ten enumerated ligands also afforded TPSA value of <140 and RBN of <12, which is in the acceptable range according to Veber’s oral bioavailability (Veber et al., 2002). Moreover, the binding energy of the top five enumerated ligands are lower than laninamivir and all the FDA-approved drugs against influenza neuraminidase thereby indicating more stable binding of the protein-inhibitor complex. Analysis of molecular properties of the top ten enumerated ligands suggested that these compounds have the potential for further improvement of both their bioavailability and binding fitness against influenza neuraminidase.

Observation of the molecular structures of the top ten enumerated ligands against influenza A suggested that the dihydropyran scaffold was mainly selected to support the chair conformation of ligands. There are 4 substitution sites surrounding this privileged structure as shown in Fig. 9. The R1 substitution site (e.g., benzamide and butyrolactone derivatives) bears the cyclic moiety that can possibly mask entrance to the active site by occupying the 150- or 430-cavity of neuraminidase as well as pointing towards the vdW1 site of the S3 subsite (i.e., R152, I222, E227) of neuraminidase. Hydroxyl substitution at the R1 site forms the carboxylic moiety that plays a major role in ligand binding via electrostatic interaction with the Elec1 site. Most of the R2 substituted moieties (e.g., pyrrolidine and amide derivatives) are heterocyclic or aromatic moieties which can exhibit van der Waals interaction with vdW1 site of the binding pocket. In contrast with other substitution site of enumerated ligands, all R3 moieties of enumerated ligands for influenza A bears the methyl group to form the acetamido moiety, which facilitates van der Waals interaction through either the vdW1 and vdW2 interaction sites. Finally, most of the R4 substituents were hydrogen atoms and therefore could not interact with the catalytic site, however, the amino moiety of R4 may possibly engage in hydrogen bonding with N347 found at the entrance of the binding pocket.

Analysis of the top ten enumerated ligands against influenza B revealed three scaffold types (e.g., dihydropyran, cyclohexene and cyclohexane rings) that support the chair conformation of the ligand inside the catalytic pocket of neuraminidase. Although most of the enumerated ligands possessed distinct scaffold (i.e., predominantly cyclohexene) but substitution sites around the privileged structure were found to be the same. The R1 substitution site of the cyclohexene scaffold (e.g., aminophenol, butyrolactone and betazole derivatives) is mainly occupied by either the aromatic ring that is attached to amino and hydroxyl groups or the heteropentacyclic ring that is surrounded by both polar and non-polar moiety as shown in Fig. 9. This site is responsible for interacting with the 150-cavity, which may possibly increase the binding affinity as can be seen in the top five enumerated NAIs against influenza B neuraminidase. The oxygen and nitrogen atoms linking the core structure and the R1 substitution site could also interact with the electrostatic side chain of amino acids in the Elec1 site. In contrast with other substitution sites, the R2 substitution site is mainly occupied by fluorine atoms, which can possibly interact with nearby positively-charged residues such as E116 and D148. R3 is a part of long non-polar chain, which facilitated van der Waals interaction with the vdW1 site inside the neuraminidase active site. Similarly, methyl and trifluoromethyl groups at the R4 substituents could facilitate either van der Waals or hydrophobic interaction with the vdW2 site and adjacent non-polar amino acids. The binding pose of A1 and B1 that afforded the lowest binding energy against neuraminidase of influenza A and B, respectively, was selected as a representative structure and shown in Figs. 10A and 10B.

Figure 10 Binding pose of enumerated ligands A1 (A) and B1 (B) providing the lowest binding energy against influenza A and B neuraminidase.

The electrostatic potential on the surface of neuraminidase is calculated via APBS and is shown by red, blue and white colors that represents negative, positive and neutral charge, respectively.

It has been shown that the evolutionary algorithm in combination with bioisosteric replacement of polar moiety as performed in AutoGrow could improve the lipophilicity of de novo designed NAIs proposed herein, which may possibly increase the percentage of bioavailability when administrated orally. Notably, the algorithm enumerated ligands targeting the 150- and 430-cavity found at the periphery of the catalytic pocket of neuraminidase. It should be noted that ligands capable of interacting with either the 150- or 430-cavity of influenza neuraminidase had the tendency of exhibiting lower binding energy. The enumerated NAIs discussed herein have been demonstrated to be promising candidates for development as drugs against influenza.

Conclusion

The emergence of novel influenza strains that possess resistance mutations emphasize the importance of finding novel therapeutic agents for treatment and prophylaxis. The increase in the emergence of influenza viruses, particularly mutant variants, calls for the development of novel promising NAIs, in addition to the three currently approved NAIs, for preparedness against influenza. Nevertheless, there are several compounds that were tested to evaluate their inhibitory activity against influenza neuraminidase. Expanding the chemical space available in public databases of NAIs provides an opportunity to investigate the molecular factors relevant to the bioactivity of NAIs. In addition, a combination of various computational approaches revealed the structure–activity relationships of NAIs, which are essential for rational drug design to develop new promising therapeutic agents against influenza neuraminidase. Therefore, this work reports a large-scale study of the chemical space of NAIs against influenza type A and B and performs statistical and QSAR investigations of both molecular and quantum chemical properties that contribute inhibitory activity against influenza neuraminidase. Moreover, maximum common molecular substructures and their functional groups were analyzed from a ligand-based perspective. In addition, the binding modes of active NAIs were investigated to observe important amino acid residues and their site-moiety preferences that facilitate protein-ligand interaction. Moreover, informative descriptors leading to good performance of the QSAR model were achieved in combination with a statistical analysis that revealed the molecular properties that distinguish between active and inactive classes of NAIs. The molecular properties of the active group include a higher number of rotatable bonds, number of hydrogen-bond donors and acceptor atoms, total energy of molecules and kinetic stability. In addition, the active group also appeared to possess fewer cyclic rings and lower lipophilicity and charge suggested by the univariate analysis. The maximum common substructures observed in NAIs are primarily cyclohexene-based, dihydropyran-based and cyclopentane-based scaffolds in the molecular framework. These fragments were suggested to be privileged structures contributing to neuraminidase inhibition. Functional group analysis revealed important functional groups and the characteristic patterns amongst active and inactive compounds. Results from the decision tree models suggested that the bioactivity of NAIs can be classified according to their functional groups. Furthermore, results from binding mode analysis revealed key interactions that facilitated protein-ligand binding along with their moiety preferences. Combinatorial library enumeration in the context of fragment-based molecular docking produced novel NAIs with higher binding fitness and robust bioavailability when compared to FDA-approved and existing lead compounds.

Supplemental Information

Supplemental Information 1 Supplementary Information

Click here for additional data file.

The authors are grateful for the insightful comments from the editor and peer reviewers for the improvement of this work. Thanks also go to Aijaz Ahmad Malik for assistance in calculating the electrostatic field of the protein using APBS for subsequent rendering in PyMOL.

Additional Information and Declarations

Competing Interests

Author Contributions

Data Availability

The authors declare there are no competing interests.

Nuttapat Anuwongcharoen and Watshara Shoombuatong performed the experiments, analyzed the data, wrote the paper, prepared figures and/or tables, reviewed drafts of the paper.

Tanawut Tantimongcolwat analyzed the data, wrote the paper, reviewed drafts of the paper.

Virapong Prachayasittikul analyzed the data, contributed reagents/materials/analysis tools, wrote the paper, reviewed drafts of the paper.

Chanin Nantasenamat conceived and designed the experiments, analyzed the data, contributed reagents/materials/analysis tools, wrote the paper, prepared figures and/or tables, reviewed drafts of the paper.

The following information was supplied regarding data availability:

Data sets of inhibitors against neuraminidase from influenza types A and B (XLSX file format) and molecular structures (SDF file format) of enumerated ligands are available as supplementary data on figshare at http://dx.doi.org/10.6084/m9.figshare.1612484.

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
