# Peer review of "Exploring the chemical space of influenza neuraminidase inhibitors"

_PeerJ, doi:10.7717/peerj.1958_

## Round 0.1 · original submission · Major Revisions

· Academic Editor

Major Revisions

Both referees believe your paper is interesting, but some effort must still be made to deem it publishable. I expect that clarification of the points raised by the reviewer #1 should not be too difficult, and I would also urge you to increase the readability of the "results" section. Specifically the series of paragraphs (p. 7-12) describing the statistical spread of the descriptor data mostly repeats the data in table 5, with hardly any gain in information but with the cost of a significant burden to the reader. Inclusion of a figure depicting binding to neuraminidase B would also be helpful.

Reviewer #2 rightly points out that experimental validation would be excellent. I deem that the work is publishable without such stringent validation, but you should provide a few examples of molecules designed according to the guidelines suggested by your research, and compare (using AutoDock VINA) their binding strength to the binding strength of known active molecules. This would allow the reader to have an additional measure of the percentage of success that might be expected in "real-life" application of the insights you have provided.

·

Basic reporting

Introduction
- Many QSAR studies of influenza neuraminidase inhibitors have been published. In this work the binary classification models were developed using decision tree. The authors should explain why the approach taken is necessarily better than the previous studies by comparing with the studies.

Materials and methods
- Specificity (Eq. 2) is TN/(TN+FP) not TN/(TP+FP). Did you use the wrong equation in this work or just type wrong equation?
- The p-value was used throughout in this work. It is necessary to explain the meaning of this value and the significance level of the test.
-Data collection: 347 and 175 NIAs for type A and B, respectively, were obtained, and the intermediate 62 and 44 NIAs were excluded. Therefore 285 and 131 NIAs for type A and B, respectively, were used for model development. However, these numbers do not agree with Table 1.

Results and discussion
- It is not readable because it is too long and has too many numbers.

Experimental design

Materials and methods
- Binding analysis: AutoDock Vina usually generates a large number of possible protein-ligand complexes. How did you select a binding conformation for the analysis?

Validity of the findings

Result and discussion
- Binding mode analysis: To further understanding of protein-ligand interaction site, docking experiments were conduced. But there is no explanation connecting the findings from developed decision tree models and docking experiments.

Reviewer 2 ·

Basic reporting

The paper correspond to all PeerJ formats

Experimental design

The paper correspond to all PeerJ formats

Validity of the findings

The paper correspond to all PeerJ formats

Additional comments

The paper is interesting, however the usefulness of structural recommendations for targeted molecular design is questionable. An example of successful experimental validation of a novel compound designed following suggested structural rules will strengthen the paper significantly.

---

## Round 0.2 · Minor Revisions

· Academic Editor

Minor Revisions

The manuscript has been improved. It is not clear, however, how your findings relating the molecular descriptors relate to the section on the combinatorial library enumeration. Were they used as input for the initial selection of "seed structures" for AutoGrow, or used to select the most promising candidates from the output of AutoGrow?

---

## Round 0.3 · Minor Revisions

· Academic Editor

Minor Revisions

The prior reviewers declined to examine your revision. But upon further reading I have a few queries which require further changes:

A) lines 154 and 155 state, like earlier versions of the paper, that the model built with the J48 algorithm used thirteen molecular descriptors. line 381, however, says that each compound was calculated as an M-vector of 307 bits, but does not state the meaning of each bit. I assume that most of these refer to the presence/absence of specific structural features. These should be given as Supporting information.
B) The performance of Bayes, ANN and SVM in Table 3 is subtly different from that shown in Table 3 from version 1. Am I correct in guessing that this change comes from the different number of descriptors used in both versions?
C) The new decision tree is not very informative: could you please change SubFPC1, etc. to a short description of the substructure fingerprint?
D) In versions 0 and 1, the plots of descriptor usage derived from decision trees showed descriptors taken from the initial set of 13 molecular characteristics (HOMO-LUMO, MW, Dipole, etc.) In version 2, the plots are very different and do not include any of these descriptors, but only substructure fingerprint descriptors. Why?
E) In version 0 the descriptors plotted in figure 5 were molecular descriptors, but the decision tree shown in figure 6 only included substructure fingerprint descriptors. Does that mean that you built two different kinds of decision trees (one using only QM- and similar properties, and another using only substructures)?
F) Structures B4 and B7 in figure 8 (version 1) seem to be B3 and B5 in version 2, but the MW/TPSA/AlogP data present in table 4 are different between both versions. Is the error in the table or in the structures?

---

## Round 0.4 · Minor Revisions

· Academic Editor

Minor Revisions

Editor's note: To avoid the delays introduced by the PeerJ editorial system, the discussion below was conducted through private email between editor and corresponding author.

Ed:

"I have carefully read your recent rebuttal and gone over the paper again, and there are still a few points where the text is not completely clear, most likely simply because English is not my (or your) mother tongue.

The y-axis of Panels C and D in Figure 4 lack legends.

lines 359-362: "Figure 4D shows that almost thirteen descriptors were considered sufficient for describing the active and inactive groups, except for RBN and MW. It should be noted that the usage of these descriptors may improve the performance of a predictive model for discriminating influenza B into the active or inactive group." Do you mean to state something like "Figure 4D shows that discriminating between the active and inactive groups requires almost all descriptors, except for RBN and MW." ?

lines 365-366 "Similarly, several descriptors were also significant for differentiating the active and inactive class of influenza B NAIs with the exception of nHDon, ALogP, energy, dipole moment and HOMO descriptors." and lines 270-372 "On the other hand, the active class of influenza B NAIs had the tendency of being smaller in size (MW), less molecular flexibility (RBN), ring moiety (nCIC), hydrogen-bond acceptors (nHAcc), charge (Qm ) but higher chemical reactivity/stability of molecules (HOMO-LUMO). " I cannot follow the reasoning, as Figure 4C seems to show no discrimination between actives and inactives for type B. Please explain.

I was surprised by the large number of chiral centers and 1,3-tautomerizable bonds (as computed by PaDel-descriptor) required for discrimination between actives/inactives in decision trees from Figure 5. As a check, I downloaded your figshare data and experimeted with the 1st molecule (SMILES string CCC(CC)OC1C=C(CC(N=C(N)N)C1NC(C)=O)P(O)(O)=O ) According to your data, PaDel Descriptor counted 10 chiral centers and 19 1,3-tautomerizable bonds, whereas only 3 chiral centers are present. I have analyzed the SMARTS cards in PaDel desctiptor, and I guess this strange behavior is due to the lack of explicit hydrogens in your SMILES strings. Could you please check the number of descriptors returned by PaDel from
[CH3][CH2][CH]([CH2][CH3])O[CH]1[CH]=C([CH2][CH](N=C(N)N)[CH]1NC([CH3])=O)P(O)(O)=O ? "

Author: "In regards to the last point on computing the PaDEL descriptors using and not using explicit hydrogens, I have tried computing the PaDEL descriptors for both forms of the suggested SMILES notation of the first molecule in PaDEL and had got identical descriptors for both (please see the attached Excel file). Thus, irrespective of having the explicit hydrogens or not, the same set of descriptors were obtained. Please kindly advise.

As for the previous 3 points, we will perform the suggested language corrections as well as add the missing panel labels."

Editor:" Since no changes are apparent in the number of structure fingerprints when explicit hydrogens are added, I think you might simply place "1,3-tautomerizable" and "chiral center specified" between "scare quotes" in the boxes in the decision tree and add a short description to the legend stating that those fingerprints correspond to idiosyncratic PaDel definitions, rather than "standard definitions".

Editor : "Please do not forget to explain the rationale behind your interpretation of PCAs for influenza B, which look (to the untrained eye) unable to discriminate between active and inactive molecules."

---

## Round 0.5 · accepted · Accept

· Academic Editor

Accept

The paper has been suitably improved and is ready for publication.